# A Convex Hull-Based Machine Learning Algorithm for Multipartite Entanglement Classification

Pingxun Wang

Department of Communications Science and Engineering, Fudan University, Shanghai 200433, China; wangpingxun0209@163.com

**Abstract:** Quantum entanglement becomes more complicated and capricious when more than two parties are involved. There have been methods for classifying some inequivalent multipartite entanglements, such as GHZ states and W states. In this paper, based on the fact that the set of all W states is convex, we approximate the convex hull by some critical points from the inside and propose a method of classification via the tangent hyperplane. To accelerate the calculation, we bring ensemble learning of machine learning into the algorithm, thus improving the accuracy of the classification.

**Keywords:** multipartite entanglement; G states; W states; ensemble learning

## 1. Introduction

Machine learning was born from pattern recognition, which possesses the ability to make decisions without explicit programming after learning from large amounts of data. Up to now, machine learning has been employed to quantum areas. Thus far, a number of promising applications have been proposed, such as quantum metric learning [1], the gate decomposition problem [2], quantum states discrimination [3], quantum discrete feature encoding [4], quantum nodes based on variational unsampling protocols [5] and quantifying steerability [6].

Entanglement was first described by Einstein, Podolsky and Rosen [7]. Later, quantum entanglement became a useful resource, enabling tasks such as quantum cryptography [8], quantum teleportation [9] and driving fields on the spectrum [10]. There are also many methods which have been proposed to distinguish and quantify entanglement, including Tsallis-q entanglement [11], device-independent entanglement witnesses [12] and the geometric measure of entanglement [13].

When it comes to the number of parties involved in entanglement, there are two typical classes: bipartite entanglement and multipartite entanglement. When there are more than two parties involved, the situation gets complicated. For example, when there are three qubits in the Hilbert space $\mathcal{H}_A$, $\mathcal{H}_B$ and $\mathcal{H}_C$, a state is called a fully separable state if it can be written as

$$\left|\Phi^{fs}\right\rangle_{A|B|C} = |\alpha\rangle_A \otimes |\beta\rangle_B \otimes |\gamma\rangle_C \tag{1}$$

where $|\alpha\rangle_A \in \mathcal{H}_A, |\beta\rangle_B \in \mathcal{H}_B, |\gamma\rangle_C \in \mathcal{H}_C$. Biseparable states can be written as a product state in the bipartite system. A biseparable state can be created if two of the three qubits are grouped together into one party. There are three possibilities for grouping two qubits together, and hence there are three classes of biseparable states. There are three possibilities: $\left|\Phi^{bs}\right\rangle_{A|BC} = |\alpha\rangle_A \otimes |\delta\rangle_{BC}$, $\left|\Phi^{bs}\right\rangle_{B|AC} = |\alpha\rangle_B \otimes |\delta\rangle_{AC}$ and $\left|\Phi^{bs}\right\rangle_{C|AB} = |\alpha\rangle_C \otimes |\delta\rangle_{AB}$, where $|\delta\rangle$ denotes a two-party state that might be entangled. Finally, a state is called genuine entangled if it is neither fully separable nor biseparable. There are two main families of multipartite entanglement: one is the GHZ states [14,15], and the other is the so-called W states [16]. A schematic picture of the structure of mixed states for three qubits is shown in Figure 1 [17].

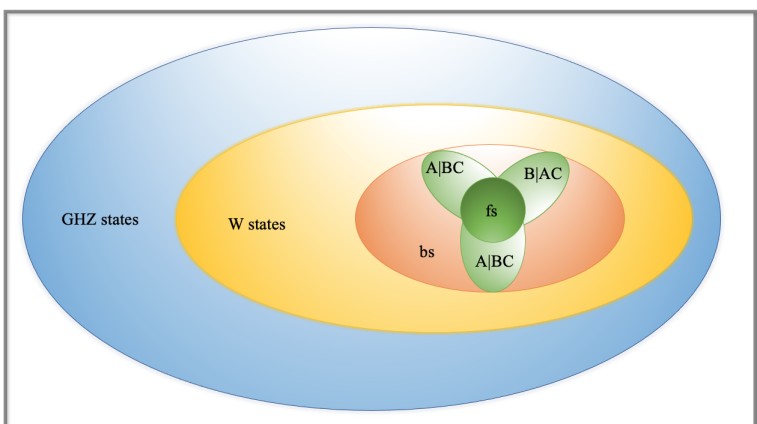

**Figure 1.** Schematic picture of the structure of mixed states for three qubits. The convex set of all fully separable states (fs) is a subset of the set of all biseparable states (bs). The biseparable states are the convex combinations of the biseparable states with respect to fixed partitions sketched by the three different leaves. Outside are the genuine tripartite entangled states, the W class and the GHZ class. There are many more GHZ states than W states. Reproduced with permission from Otfried Gühne, Géza Tóth, Physics Reports; published by Elsevier, 2009.

Given two three-qubit states $|\phi\rangle$ and $|\psi\rangle$, one can ask whether it is possible to transform a single copy of $|\phi\rangle$ into $|\psi\rangle$ with local operations and classical communication (LOCC) without requiring that this be done with certainty. This operation is called stochastic local operations and classical communication (SLOCC). Compared with the well-known local operations and classical communication (LOCC), SLOCC has a non-unit probability. For systems of $N$ qudits described by Hilbert spaces of the form $\mathcal{H} := \mathbb{C}^{d_1} \otimes \ldots \otimes \mathbb{C}^{d_N}$, SLOCC operations are mathematically described by the group $G := SL(d_1, \mathbb{C}) \times \ldots \times SL(d_N, \mathbb{C})$, and the action is given by the tensor product. We call two states equivalent if there is a non-vanishing probability of success when trying to convert one to another through SLOCC. The distinction between a GHZ state and a W state is that one cannot be transferred to another through SLOCC [18]. In that case, we can establish an equivalence relation stating that two states $|\phi\rangle$ and $|\psi\rangle$ are equivalent if the parties have a no probability of success when trying to convert $|\phi\rangle$ into $|\psi\rangle$ and also $|\psi\rangle$ into $|\phi\rangle$ when $|\phi\rangle$ is a GHZ state and $|\psi\rangle$ is a W state. However, this conversion can happen when both states are GHZ or W states. This relation has been termed stochastic equivalence. Their equivalence under SLOCC indicates that both states are again suited to implement the same tasks of QIT, although this time, the probability of a successful performance of the task may differ from $|\phi\rangle$ to $|\psi\rangle$. For instance, in a three-qubit case, we have that $|\psi\rangle$ can be locally converted into $|\phi\rangle$ if an operator $A \otimes B \otimes C$ exists, satisfying

$$|\phi\rangle = A \otimes B \otimes C |\psi\rangle \tag{2}$$

where operator $A$ contains contributions from any round in which party A takes action on its subsystem and likewise for operators $B$ and $C$. To make sure the opposite conversion can happen, each of these operators is necessarily invertible, particularly such that

$$|\phi\rangle = A^{-1} \otimes B^{-1} \otimes C^{-1} |\psi\rangle \tag{3}$$

There has been an abundance of methods for classifying separable and entangled states. For instance, for $2 \times 2$ and $2 \times 3$ systems, the PPT criterion is a well-known method [19], the computable cross-norm or realignment (CCNR) criterion is simple and strong [20,21], and the entanglement witness is a necessary and sufficient criterion in terms of directly measurable observables [22]. However, criteria for classifying GHZ and W states are relatively few at present. In this work, we employ machine learning techniques to tackle

the GHZ and W states by recasting them as learning tasks. Namely, we attempt to construct a GHZ-W classifier. Our idea is to give the classifier a large number of sampled trial states and corresponding category labels and then train the classifier to predict the category labels of new states that it has not encountered before.

## 2. The Construction of the Classifier

### 2.1. Convex Hull Approximation

As with the detection of entanglement, a natural question is asked: how is it decided which class a given state belongs to? However, methods for distinguishing between the GHZ class and the W class are very rare. For the detection of entanglement, the entanglement witness [23] can be used, as the W class states form a convex set. However, it is not clear how one can show that a state is tripartite entangled and belongs to the W class. This cannot be accomplished with witnesses; since they are designed to show that a state lies outside a convex set, they fail to prove that a state is inside a convex set. The traditional entanglement witness is used as a separability-entanglement classifier. An observable $\mathcal{W}$ is called an entanglement witness (or witness for short) if $Tr(\mathcal{W}\varrho_s) \geq 0$ for all separable $\varrho_s$ and $Tr(\mathcal{W}\varrho_e) < 0$ for at least one entangled $\varrho_e$ holds. Thus, if one measures $Tr(\mathcal{W}\varrho) < 0$, then one knows for sure that the state $\varrho$ is entangled. From a geometric point of view, both the state spaces and the entangled spaces form a convex set. The witness forms a hyperplane in the space, dividing it into two parts.

However, the boundaries of the convex set are so complicated that direct application of supervised learning to the classification will not be satisfying. In addition, due to the lack of prior knowledge for training, the neural network cannot provide an acceptable accuracy. In a study in 2018, researchers proposed a method of classification for entangled states and separable states via constructing a convex hull approximating the set of entangled states [24]. Entangled states form a convex set in the spaces, and as shown in Figure 1, W states also form a convex set inside the set of GHZ states. Therefore, inspired by [24], we introduce the convex hull approximation [25] here. The construction of convex hull is one of the most fundamental problems in computational geometry. Here, we approximate the W states from inside, for the W state space is a close convex set, and its critical points are all the pure W states [17]. We define a convex hull as follows:

$$\mathcal{C} = conv(\{c_1, \ldots, c_n\}) \tag{4}$$

where $c_1, \ldots, c_n \in \mathcal{X}$ are pure W states sampled randomly. $\mathcal{C}$ is said to be a convex hull approximation (CHA) of the W state space. To find the critical points, we propose an iterative algorithm in Section 3.2.2. With the increasing of the number of critical points, the convex hull approaches the W state space. In other words, $\mathcal{C}$ will be a more accurate CHA of the W state space if we construct it with more pure W states. With this we can approximately tell whether a state $\rho$ is a W state or not by testing if its feature vector is in $\mathcal{C}$. This is equivalent to determining whether the feature vector can be written as a convex combination of $c_i$ by solving the following linear programming:

$$max \ \alpha \quad s.t. \ \alpha p \in \mathcal{C} \tag{5}$$

The constraint condition is equivalent to the following expansion:

$$\alpha p = \sum_{i=1}^{m} \lambda_i c_i, \quad \lambda_i \geq 0, \quad \sum_i \lambda_i = 1 \tag{6}$$

Here, $\alpha = \alpha(\mathcal{C}, p)$ is a function of $\mathcal{C}$ and $p$, and $p$ is the feature vector of the state $\rho$ to be tested. If $\alpha(\mathcal{C}, p) \geq 1$, then $p$ is in $\mathcal{C}$, and thus $\rho$ is a W state; otherwise, $\rho$ it is highly possible that this is a GHZ state. A schematic picture of the convex hull approximation is shown in Figure 2.

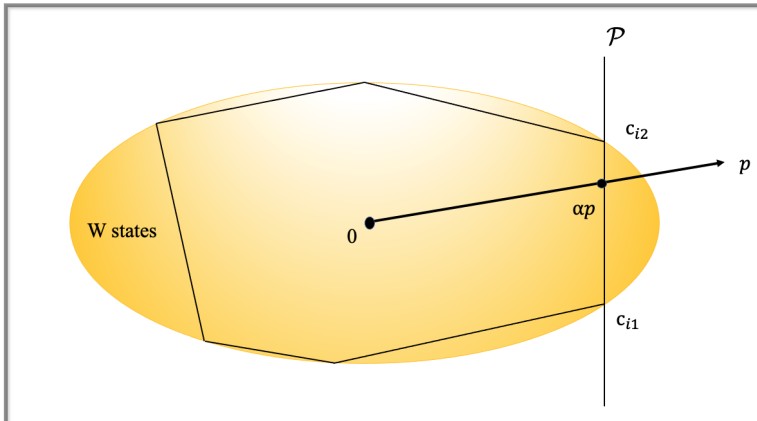

**Figure 2.** Convex hull approximation. The more critical points there are, the better the approximation is.

Then, we propose an iterative algorithm for detecting the W states. At the first step, we use a few critical points (i.e., pure W states to build a CHA $\mathcal{C}$). For a state $\rho$ whose feature vector is $p$, we find the maximum $\alpha$ by Equation (5), and it is still in $\mathcal{C}$. If $\alpha \geq 1$, then $\rho$ is certainly in CHA and thus is a W state. Otherwise, suppose $\alpha p$ lies on a hyperplane $P$ such that $P \cap C$ is the boundary of $\mathcal{C}$. We can enlarge $\mathcal{C}$ by sampling the pure W states near the known critical points. Then, we repeat the above procedure many times until $\alpha \geq 1$ or $\alpha$ converge.

### 2.2. A Tangent-Based Classifier via CHA

Due to the fact that the set of W states is convex, there must exist many tangent hyperplanes $\mathcal{P}$ so that all the W states are on the same side of the space divided by the hyperplane (i.e., $N(\mathcal{P})|\rho\rangle_W > 0$ is satisfied with all W states, where $N(\mathcal{P})$ is the normal vector of the tangent hyperplanes $\mathcal{P}$).

A single tangent hyperplane $\mathcal{P}$ is not enough to classify the W and GHZ states, but with enough tangent hyperplanes, the error rate can be tolerated. Thus, we aim to generate enough tangent hyperplanes here. Due to the fact that the boundary of the set of W states $\mathcal{W}$ is so complicated, it is difficult to find tangent hyperplanes directly. However, with enough critical points on the convex hull, we could approximate the tangent hyperplane by a hyperplane determined by some points on the convex hull that are close enough (i.e., the volume of the formed hyper-body would be minimized). With $n$ dimensions for the states, $n$ critical points are needed here.

Suppose we have a set $\mathbf{P} = \{\mathcal{P}_1, \ldots \mathcal{P}_k\}$ now, so $|\rho\rangle$ being a W state approximately equals to $N(\mathcal{P}_i)|\rho\rangle > 0$, holding for $i \in \{1, \ldots, k\}$. To approximate $\mathcal{P}$, we implement the method of Voronoi diagrams [26].

#### 2.2.1. Voronoi Diagram

A Voronoi diagram is a given finite set of points that divides the space into some small regions based on the nearest neighbor principle, where the points in the region are closer to the points in the set of points contained in the region than to any other store in the set of points. Given a set $S = \{p_1, p_2, \ldots, p_n\}$ constructed by n points, the Voronoi district of $p_i$ comes to $Reg(p_i) = \{p|d(p, p_i) < d(p, p_j), i \neq j\}$, where $d(p_i, p_j)$ refers to the Minkowski distance between $p_i$ and $p_j$. The division given by $Reg(p_i)(i = 1, 2, \ldots, n)$ and their boundary is called the Voronoi diagram generated by $p_i$.

#### 2.2.2. Minimum Hyper-Body

As was mentioned above, to find the most approximately tangent hyperplanes means to find the minimum hyper-body. Here, we utilize critical points and their neighbors to generate the Voronoi diagram, and it was proven in [26] that the optimal time complexity is $O(nlogn)$. We implement the method of divide-and-conquer. The intersec-

tion of the region where the target point is located is called the vertex. In a Voronoi diagram, the existence of a vertex implies the existence of a hyper-body. Figure 3 is a plane Voronoi diagram schematic. $q_1$, $q_2$ and $q_3$ are three arbitrary points in the given $n$ points. Point $O$ is the barycenter of the triangle constructed by $q_1$, $q_2$ and $q_3$, which means that there are no other target points in the triangle. Therefore, we can approximately consider the hyper-body composed of adjacent target points to be a locally optimal solution. In the plane Voronoi diagram case, we first select segment $q_1q_2$ as a side of the triangle and then divide the $n - 2$ points other than $q_1, q_2$ into two subsets: $S_1 = \{p | Target\ points\ in\ the\ areas\ adjacent\ to\ the\ area\ where\ q_1\ and\ q_2\ are\ located\}$ and $S_2 = \{p | Target\ points\ in\ other\ areas\}$. We can select $q_3$ and $q$ as the present, so we obtain $\triangle q_1q_2q_3$ and $\triangle q_1q_2q$. Therefore, we just have to select all triangles made by the Voronoi vertex and compare them to find the triangle with the smallest area as the solution.

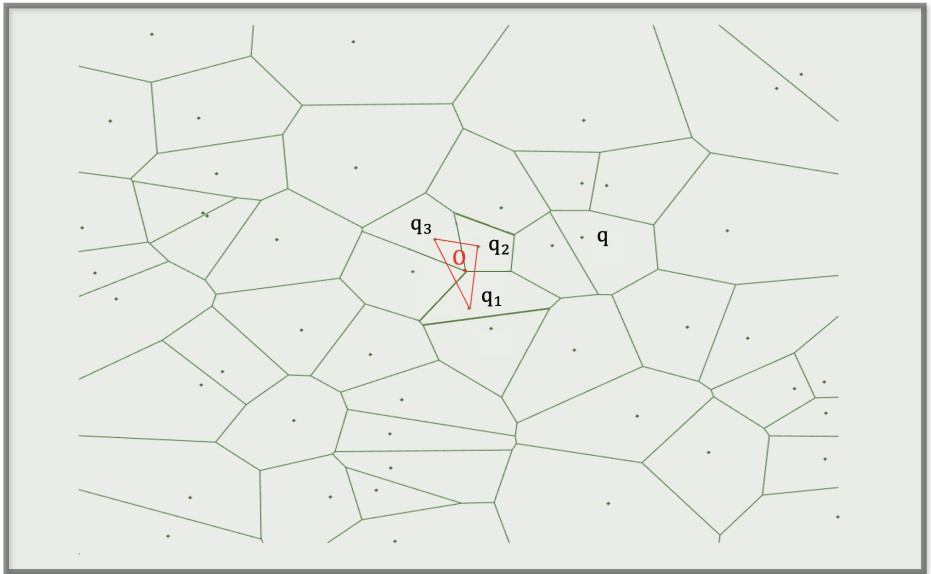

**Figure 3.** A plane Voronoi diagram schematic. $q_1$, $q_2$ and $q_3$ are three arbitrary points in the given $n$ points. Point $O$ is the barycenter of $\triangle q_1q_2q_3$. In a Voronoi diagram, there is a property where the vertex point is the barycenter of the triangle formed by the target points. Therefore, there will not be any other target points in $\triangle q_1q_2q_3$. Point $O$ is an arbitrary target point, so it must be outside of the triangle. In this case, $\triangle q_1q_2q_3$ can be considered the minimum triangle.

### 2.3. Combining CHA and Machine Learning

The method above (see Section 2.1) can detect the GHZ states and W states. However, when increasing the accuracy, we have to enlarge the convex hull so that there is a large number of critical points waiting for determination of whether they are in the convex hull or not, leading to a greater time cost. Therefore, we bring supervised learning here to speed up the algorithm.

#### 2.3.1. Data Preparation

As for the data preparation, for any quantum state $\rho$, the density operator acting on $\mathcal{H}_A \otimes \mathcal{H}_B \otimes \mathcal{H}_C$ can be presented as a real vector in $\mathcal{X} = \mathbb{R}^{d_A^2 d_B^2 d_C^2 - 1}$ due to the fact that $\rho$ is Hermitian and of trace 1, where $d_i$ is the dimension of $\mathcal{H}_i$. Generalized Gell-Mann matrices [27] are a frequently used linear independent Hermitian orthonormal basis here.

Let $\{|1\rangle, \ldots, |n\rangle\}$ be the computational basis of the n-dimensional Hilbert space and

$$E_{j,k} = |j\rangle\langle k| \tag{7}$$

Thus, there are symmetrical matrices presented as follows:

$$S_{j,k} = E_{j,k} + E_{k,j}, \ 1 \leq j \leq k \leq n \tag{8}$$

There are also anti-symmetrical matrices, presented as

$$A_{j,k} = -i\left(E_{j,k} - E_{k,j}\right), \ 1 \leq j \leq k \leq n \tag{9}$$

and diagonal matrices, presented as

$$D_l = \sqrt{\frac{2}{l(l+1)}} \left( \sum_{j=1}^{l} E_{j,j} - lE_{l+1,l+1} \right), \ 1 \leq l \leq n-1 \tag{10}$$

Therefore, the generalized Gell-Mann matrices can be presented as follows:

$$\{\lambda_i\} = \left\{ S_{j,k} \right\} \cup \left\{ A_{j,k} \right\} \cup \{D_l\} \tag{11}$$

where $tr(\lambda_i) = 0$ and $tr(\lambda_i \lambda_j) = 2\delta_{i,j}$ when $i \neq j$.

Therefore, every $\rho$ can be expanded into linear combinations as follows:

$$\rho = \frac{1}{n}\left( \mathbb{I} + \sqrt{\frac{n(n+1)}{2}} x \cdot \vec{\lambda} \right) \tag{12}$$

where $x_i = \sqrt{\frac{n}{2(n-1)}} tr(\rho \lambda_i)$.

In supervised learning, the training set should have the following form:

$$D_{train} = \{(x_1, label_1), \ldots, (x_m, label_m)\} \tag{13}$$

where $m$ is the size of the training set, $x_i$ is the state vector input and $label_i$ is the corresponding tag such that $label_i \in \{0, 1\}$.

### 2.3.2. Extended Data Form

The CHA method described above has another obvious drawback from the perspective of the trade-off between accuracy and time consumption. Improving the accuracy means adding additional critical points to expand the convex hull, which leads to a greater time cost to determine whether a point is within the expanded convex hull. To overcome this problem, we combine CHA with supervised learning, as machine learning has the ability to speed up this computation. To boost the accuracy, we add more information into Equation (13). The original feature vectors $x$ are extended to $(x, \alpha)$ so it can contain the boundary information. Therefore, the dataset can be rewritten as

$$\mathcal{D}_{train} = \{(x_1, \alpha_1, label_1), \ldots, (x_n, \alpha_n, label_n)\} \tag{14}$$

where $\alpha_i = \alpha(\mathcal{C}, x_i)$. Therefore, the classifier $h$ is defined on $\mathbb{X} \times \mathbb{R}$. The loss function of $h$ is defined as follows:

$$L(h, \mathcal{D}_{train}) = \frac{1}{|\mathcal{D}_{train}|} \sum_{(x_i, \alpha_i, label_i) \in \mathcal{D}_{train}} \mathbb{I}(y_i \neq h(x_i, \alpha_i)) \tag{15}$$

Then, we can employ a standard ensemble learning approach to train a classifier.

### 2.3.3. Ensemble Learning

In supervised learning algorithms, the goal is to learn a stable model that performs well in all aspects, but in practice, this is often not the case, and sometimes, we can only obtain multiple models with preferences. The underlying idea of integration learning is that even if a weak classifier gets a wrong prediction, the other weak classifiers can correct the

error [28]. In the bagging method, the bootstrap method is used to obtain N datasets from the overall dataset with put-back sampling, a model is learned on each dataset, and the final prediction results are obtained using the output of the N models. Specifically, the classification problem uses the N model prediction voting method, and the regression problem uses the N model prediction averaging method. Boosting is a machine learning algorithm that can be used to reduce bias in supervised learning. It is also mainly used to learn a series of weak classifiers and combine them into a strong classifier. Each training example is assigned an equal weight at the beginning of training, and then the algorithm is used to train the training set for several rounds. After each training, the training examples that fail are assigned a larger weight (i.e., the learning algorithm pays more attention to the wrong samples after each learning, resulting in multiple prediction functions). Here, we imply both two models to compare their effects.

## 3. The Performance of the Classifier

### 3.1. Training Phase of the Predictors

Here, we generated GHZ and W states directly, calling the functions GHZState.m and WState.m as in [29]. We generated a specific number of random quantum states which were either GHZ states or W states with different labels. Then, we generated SLOCC to transfer these states and obtain the training set, as is shown in Figure 4.

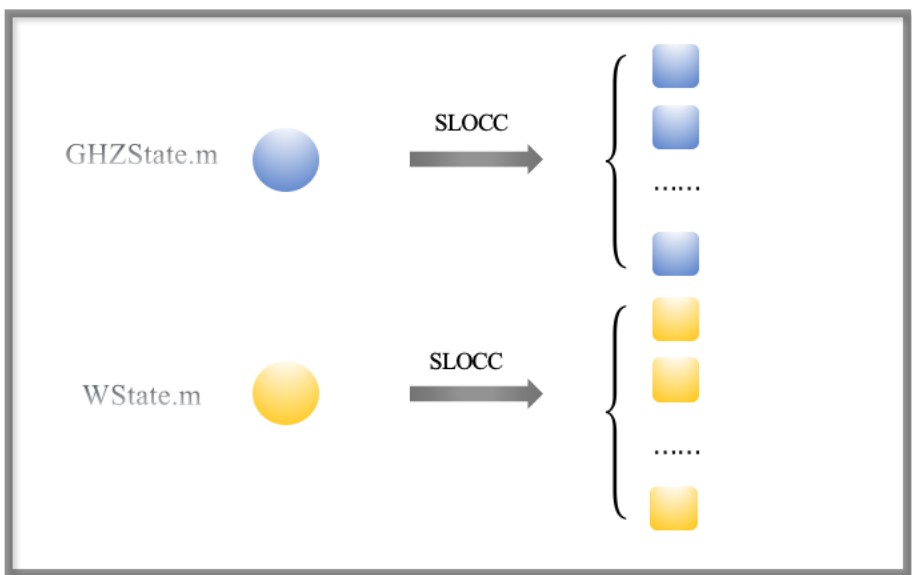

**Figure 4.** Schematic diagram for generating the training set. We implemented the typical forms of GHZ states $|GHZ\rangle = \frac{|0\rangle^{\otimes n} + |1\rangle^{\otimes n}}{\sqrt{2}}$ and W states $|W\rangle = \frac{1}{\sqrt{n}}(|100...0\rangle + |010...0\rangle + ... + |000...1\rangle)$ as the original sates and generated SLOCC to transfer 191 these states and form more states in the GHZ states set and W states set. We adopted the result as the training set.

### 3.2. Testing Phase of the Predictors

With the training set, we could construct the convex hull. Here, we choose an iterative algorithm to find more critical points by one known critical point (i.e., to find its neighbors). The algorithms are shown below.

3.2.1. Algorithm for Calculating the $\alpha$

Let the GHZ states $|\phi\rangle_{GHZ} \in \mathcal{H}_A$ and the W states $|\psi\rangle_W \in \mathcal{H}_B$. To approximate the set of W states with a convex hull $C$, we generated a bunch of critical points. The process was carried out as follows:

(1)　Randomly sample a state $|\phi\rangle_{GHZ} \in \mathcal{H}_A \cong \mathbb{C}^{d_A}$ from a uniform distribution according to the Haar measure;

(2)     Randomly sample a state $|\psi\rangle_W \in \mathcal{H}_B \cong \mathbb{C}^{d_B}$ from a uniform distribution according to the Haar measure;

(3)     Return $|\phi\rangle_{GHZ}|\psi\rangle_W$.

Execute the process above N times to obtain $n$ critical points $c_1, \ldots, c_n$. Then, solve the convex optimization problem mentioned above to decide whether the vector is in the convex hull generated.

### 3.2.2. Algorithm for Finding Critical Points

For the set of W states $\mathcal{W}$, which is closed and convex, for an arbitrary state $\rho$, there must exist and only exist a critical point satisfying that $\alpha_\rho \rho + (1 - \alpha_\rho)I/(d_A d_B)$ is on the boundary of $\mathcal{W}$. When $\alpha \leq \alpha_\rho$, $\rho$ is a W state, and when $\alpha > \alpha_\rho$, it is a GHZ state. Here is an iterative algorithm for calculating $\alpha_\rho$ based on the convex hull $\mathcal{C}$:

(1)     Initiate $p$ as the feature vector of $\rho$, and set $\varepsilon = 1, \xi = 0.9$;

(2)     Update $\alpha_\rho \leftarrow \alpha(\mathcal{C}, p)$;

(3)     Now, $\alpha_\rho p = \sum_i \lambda_i c_i$. Pick $c_{i_1}, \ldots, c_{i_D}$ to be the critical points satisfying $\lambda_{i_k} > 0$. Update $\mathcal{C} \leftarrow conv(c_{i_1}, \ldots, c_{i_D})$;

(4)     For each $k = 1, \ldots D$, suppose $c_{i_k}$ is the feature vector of $|a_k\rangle|b_k\rangle$. Sample the neighbor of $c_{i_k}$; that is, to randomly generate two Hermitian operators $H_1 \in \mathcal{H}_A, H_2 \in \mathcal{H}_B$, satisfying $\|H_1\| = 1, \|H_2\| = 1$. Let $\delta$ be a random number in $[0, \varepsilon]$. Set $|a_k'\rangle|b_k'\rangle = (e^{i\delta H_1} \otimes e^{i\delta H_2})|a_k\rangle|b_k\rangle$. Set $\mathcal{C} \leftarrow conv(\mathcal{C}, c_k')$, where $c_k'$ is the feature vector of $|a_k'\rangle|b_k'\rangle$;

(5)     Update $\varepsilon \leftarrow \xi\varepsilon$ and go back to step 2.

We repeated step 4 10 times to find enough neighbors. The initiation of $\xi$ could be adjusted. When $\xi$ is closer to 1, the approximation is more precise, and on the contrary, when $\xi$ is closer to 0, the speed of approximation gets faster.

### 3.2.3. Algorithm for Calculating an Approximate Tangent Hyperplane

For a given set of critical points $S = \{p_1, p_2, \ldots, p_n\}$, we chose a Voronoi diagram to find the approximate tangent hyperplanes:

(1)     Divide the $n$ critical points into 50 parts, each of which contains $m$ critical points. Generate a Voronoi diagram via each part. Here, we directly implement the function voronoin.m of the Qhull toolbox [30] to generate the $n$-dimensional Voronoi diagram.

(2)     Find the minimum hyper-body via the adjacent target points. Generate the corresponding tangent hyperplane. Decide which states are GHZ states according to the hyperplane.

(3)     Repeat step 2 50 times or until all the diagrams have been used.

Without the implementation of CHA, the results of the direct supervised learning are shown below (Table 1).

**Table 1.** Error rate of the classifier via the direct supervised learning algorithm.

| Method | SVM | Decision Tree | Bagging | Boosting |
|---|---|---|---|---|
| Error (%) | 14.1 | 25.2 | 18.8 | 17.3 |

Here, we name the CHA combined with a tangent hyperplane as TCHA and name the CHA combined with ensemble learning as ECHA. For the lack of boundary knowledge, it was difficult to reduce the error rate. For the three-qubit and four-qubit cases, the error rates of different numbers of critical points are shown below (Figures 5 and 6).

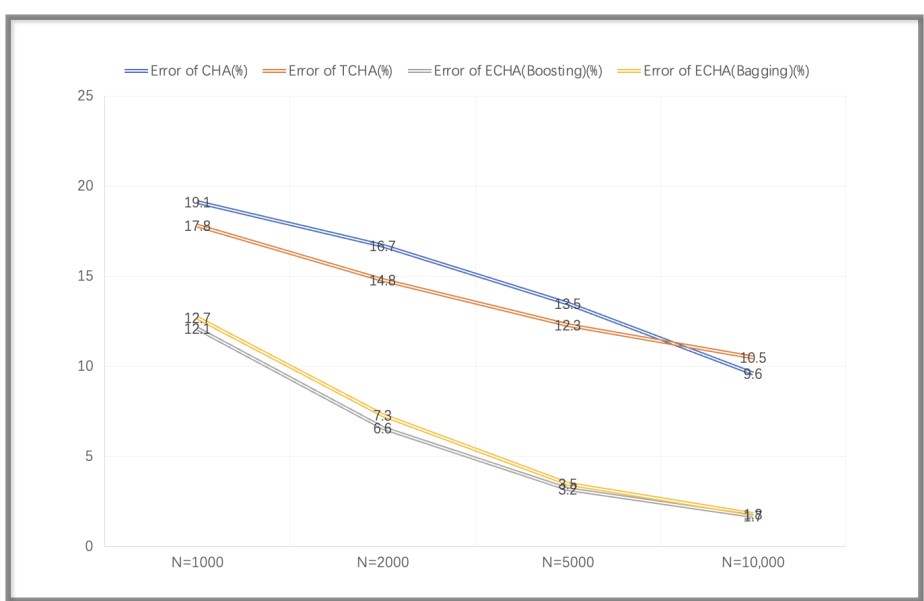

**Figure 5.** The error rates of different classifiers for the 3-qubit case when N increased. The performances of the two ECHAs were better than the CHA, while those of the ECHAs were similar.

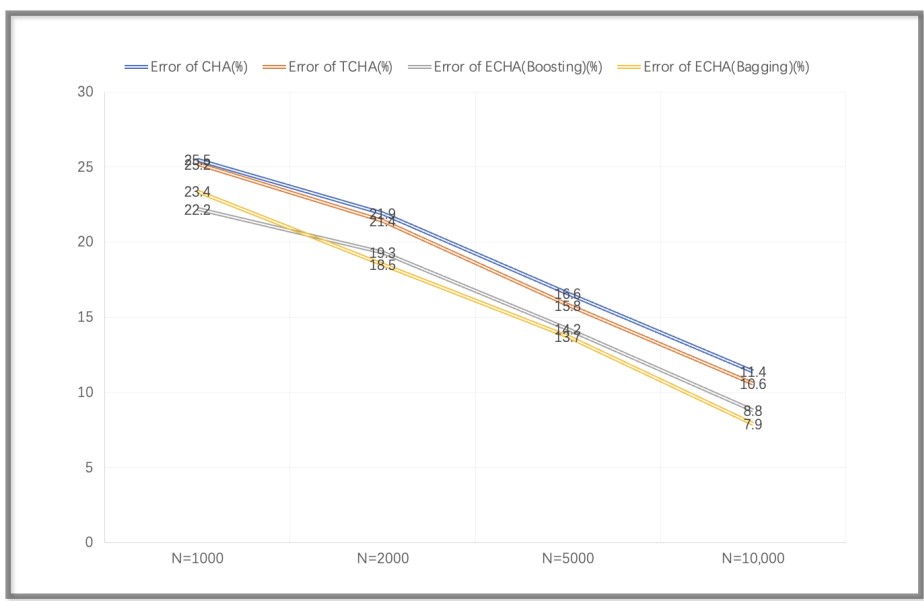

**Figure 6.** The error rates of different classifiers for the 4-qubit case when N increased. They were slightly poorer than those for the 3-qubit case.

It can be seen that the accuracy decreased when the number of critical points increased. The performances of the three-qubit cases were better than those of the four-qubit cases. The approximation was more accurate when the number of critical points increased, and when the number of qubits increased, the entanglements became more complicated. For the case of the TCHA, the performance was better than that of the CHA due to the introduction of a tagent hyperplane. However, TCHA highly depends on the number of critical points, leading to a huge amount of computations, so the performances have limits. The performances of the ECHA were better than those of TCHA and CHA. The training of the machine learning model is related to the dimensions of the feature vectors, which was 65 in the case of the 3-qubit, case and increased to 257 in the case of the 4-qubit case. The dimensions of the feature vectors predictably grew as the number of qubits grew, so the accuracy would be reduced.

## 4. Conclusions

In this paper, we built a GHZ-W state classifier by an ensemble learning approach. To improve the accuracy,we first implemented a Voronoi diagram to build a tangent hyperplane classifier. Then, we added the boundary information about the convex hull as prior knowledge in the data to be trained and tested it to build an ensemble learning classifier. Such classifiers outperformed the algorithm with direct supervised learning in terms of accuracy.

The key of our scheme is the approximation of the convex hull of quantum states, so in theory, this method can be implemented with other classifications of multipartite entanglements meeting the conditions of a convex set. For example, the hierarchy of multipartite entangled states among $N$-party quantum states meets this condition, so genuine multipartite entangled states and $k$ separable states can be classified via this method. Classifications of W states as well as Dicke states, cluster states and graph states are part of the same case [31]. We hope our scheme can be implemented in other types of entanglement classification. Aside from that, theoretically, such a classifier can also be extended to higher dimensions. We hope that our classifier will be able to handle more quantum information tasks in the future.

**Funding:** This research received no external funding.

**Institutional Review Board Statement:** Not applicable.

**Informed Consent Statement:** Not applicable.

**Data Availability Statement:** The data presented in this study are available at https://github.com/Pingxun-Wang/GHZ-W-states-classifier, accessed on 1 March 2022.

**Acknowledgments:** The authors would like to thank S. Qian of the Communication Science and Engineering Department of Fudan University for helpful discussions on topics related to this work. The authors are grateful to J. Ren, J. Xu and J. Zhan for help with proofreading and other discussions.

**Conflicts of Interest:** The author declares no conflict of interest.

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
