# Peer review of "A Convex Hull-Based Machine Learning Algorithm for Multipartite Entanglement Classification"

_applsci, doi:10.3390/app122412778_

Round 1
Reviewer 1 Report
This manuscript investigates the state classification based on machine learning. I have the following questions/comments before suggesting its acceptance.
In line 52, the paper said ‘classifying GHZ and W states are needy at present.' The concern/question is how this “needy” problem influences the supervised learning algorithm? Will this hamper the generation of labeled data, in terms of speed and data set size?
In line 81, can you comment more on how the critical points are prepared [especially considering the above question]?
In Eq-(5) and (6), and the following paragraph Line 88-90, the “p” is not defined.
In line-38, can you comment more on the state transformation based on SLOCC? Especially why the ‘stochastic’ is important here [instead of the well-known LOCC]?
In line-51, please list several references of ‘classifying separable and entangled states’.
In line-67, is the observable W not W state [and next line it becomes \mathcal{W}]? Or a symbol conflict?
In Fig-3, why O is the vertex of the triangle constructed by q1~q3? The “O” point looks like the tiny blue dot below the red triangle.
Author Response
Dear reviewer,
Thank you for your time to review this manuscript. We feel great thanks for your professional work on our article. As you are concerned, there are several problems that need to be addressed. Here are our notes to your nice comments:
Q1: In line 52, the paper said ‘classifying GHZ and W states are needy at present.' The concern/question is how this “needy” problem influences the supervised learning algorithm? Will this hamper the generation of labeled data, in terms of speed and data set size?
A1: I’m sorry that the English expression went wrong here. What we tried to express is that there are few classifying methods now, so the right expression should be “There has been abundant methods of classifying separable and entangled states. However, that of classifying GHZ and W states are relatively few at present.” We have modified this problem in the manuscript in the new version.
Q2: In line 81, can you comment more on how the critical points are prepared [especially considering the above question]?
A2: In chapter 3.2.2, we proposed an iterative algorithm for finding critical points. The key step is to solve a convex function in step 3). We have added the comment in the new version.
Q3: In Eq-(5) and (6), and the following paragraph Line 88-90, the “p” is not defined.
A3: I’m sorry for our oversight. “p” is the feature vector of the state to be tested. We have added the comment in the new version.
Q4: In line-38, can you comment more on the state transformation based on SLOCC? Especially why the ‘stochastic’ is important here [instead of the well-known LOCC]?
A4: The ‘stochastic’ means a group of operations. Compared to LOCC, SLOCC has a non-unit probability. We have added the comment in the new version.
Q5: In line-51, please list several references of ‘classifying separable and entangled states’.
A5: We have added the well-known PPT criterion, CCNR criterion and entanglement witness criterion in the new version.
Q6: In line-67, is the observable W not W state [and next line it becomes \mathcal{W}]? Or a symbol conflict?
A6: I’m sorry for the symbol problem. We have corrected it as \mathcal{W} in the new version.
Q7: In Fig-3, why O is the vertex of the triangle constructed by q1~q3? The “O” point looks like the tiny blue dot below the red triangle.
A7: We’re sorry there are some mistakes and unclear presentations in the figure and caption. Point O is the barycenter of the triangle. We have drawn a new figure and improve the caption to explain this method more clearly.
Thank you again for your positive comments and valuable suggestions to improve the quality of our manuscript. If there are any other modifications we could make, we would like very much to modify them and we really appreciate your help. Thank you very much for your help.
Reviewer 2 Report
The manuscript proposes a scheme for classifying the entanglements, employing a machine learning algorithm. Recently, research on the application of machine learning to quantum information processing has attracted more attention for NISQ, such as classifying quantum states. The authors has studies the problem for classifying two types of entanglements, e.g. GHZ state and W state, which are typical resource for fundamental quantum information processing, two types of entanglements. Overall, this manuscript is clear, and the technical content is understandable. Although their method is limited to GHZ state and W state, the present manuscript will provide meaningful motivation to both theorists and experimentalists for the application of machine learning to quantum information processing. Thus, I recommend the publication of this manuscript after the following optional (minor) changes.
1. The authors mention Ref.[19]. I suggest that the authors should explain in a little more detail the difference between the proposed method and the method introduced in Ref.[19].
2. The order of Ref.[25] is not correct.
3. In Fig.3, it would be helpful to explain q as well in the caption.
4. I have not understood Figure 4. It would be helpful if the authors could describe the training set in the caption a little more.
5. The authors consider only the case for GHZ state and W state, while it may be theoretically worth describing the applicability of their scheme to other types of entanglement, for example, the cluster state.
6. It needs to be modified about line breaks, for example, lines 48 and 149-151 should not be new lines.
Author Response
Dear reviewer,
Thank you for your time to review this manuscript. We feel great thanks for your professional work on our article. As you are concerned, there are several problems that need to be addressed. Here are our notes to your nice comments:
Q1: The authors mention Ref.[19]. I suggest that the authors should explain in a little more detail the difference between the proposed method and the method introduced in Ref.[19].
A1: In Ref[19], researchers proposed a method of classification of entangled states and separable states via constructing a convex hull approximating the set of entangled states. We found that the W states set and GHZ states can also be approximated in the similar way. We have added more detail of the difference in the new version.
Q2: The order of Ref.[25] is not correct.
A2: We're sorry for our oversight. We have checked the order of all references and corrected it in the new version.
Q3: In Fig.3, it would be helpful to explain q as well in the caption.
A3: We have drawn a new figure and improve the caption to explain this method more clearly.
Q4: I have not understood Figure 4. It would be helpful if the authors could describe the training set in the caption a little more.
A4: We have added a little more detail to explain the process in the new version.
Q5: The authors consider only the case for GHZ state and W state, while it may be theoretically worth describing the applicability of their scheme to other types of entanglement, for example, the cluster state.
A5: Thank you for your advice. The key of our scheme is the convex set of quantum states, so theoretically speaking, this method can be implemented in any eligible classification. Besides GHZ states and W states, there are other multipartite entangled states including Dicke states, graph states, cluster states and so on. We have added some relative discussions and references in the conclusion chapter. Due to the time limit, we haven’t performed simulation experiments, and we will continue this research in the future.
A6: It needs to be modified about line breaks, for example, lines 48 and 149-151 should not be new lines.
Q6: We're sorry for our oversight. We have checked indents and line breaks in the manuscript and modified them in the new version.
Thank you again for your positive comments and valuable suggestions to improve the quality of our manuscript. If there are any other modifications we could make, we would like very much to modify them and we really appreciate your help. Thank you very much for your help.
Round 2
Reviewer 1 Report
My questions are nicely addressed, thank you.
Reviewer 2 Report
I have no further comments than what can be found in my previous review report. I appreciate the authors have taken my suggestions into account in their revised manuscript, and recommend publication in Applied Sciences.